# Tailoring digital apps to support active ageing in a low income community

**Paula Costa Castro**[1]☯*, **Lua Bonadio Romano**[1]☯, **David Frohlich**[2]☯, **Lorena Jorge Lorenzi**[1]☯, **Lucas Bueno Campos**[1]☯, **Andresa Paixão**[1]☯, **Patrícia Bet**[3]☯, **Marije Deutekom**[4]☯, **Ben Krose**[5]☯, **Victor Zuniga Dourado**[6]☯, **Grace Angélica de Oliveira Gomes**[1]☯

**1** DGero, Universidade Federal de São Carlos, São Carlos, SP, Brazil, **2** DWRC, University of Surrey, Guildford, United Kingdom, **3** Programa de Pós-Graduação Interunidades em Bioengenharia (EESC/FMRP/IQSC-USP), Sao Carlos, SP Brazil, **4** Amsterdam University of Applied Science, Inholland University, Diemen, Holland, Netherland, **5** UVA, HVA, Amsterdam, Holland, Netherland, **6** DCMH, Universidade Federal de São Paulo, Santos, SP, Brazil

☯ These authors contributed equally to this work.
* castro@ufscar.br

**Data Availability Statement:** All relevant data are within the manuscript and its Supporting information files.

**Funding:** L. B. R. received from the funder São Paulo Research Foundation (FAPESP) N˚ 18/

## Abstract

Despite physical activity being one of the determinants of healthy aging, older people tend to become less active over the years. Maintaining physical activity levels during the life course is a motivational challenge. Digital tools have been used to change this pattern, such as smartphone applications to support physical activity; but there is a lack of in-depth research on the diversity of user's experiences, especially considering older users or non-users of information and communication technologies. Objective: Our goal was to identify requirements for designing a mobile app to encourage physical activity in a low-income community population of older people in Brazil (i.e. over 40 years old). Method: We conducted a qualitative focus group study, involving by co-design of a physical activity application (Pacer)®. Seventeen volunteers were divided into 2 focus groups of physical active and insufficiently active, and 2 further 4 subgroups in each characterised by digital engagement. The following procedures were performed: (i) baseline assessments; (ii) a focus group with physically active older people and a focus group with insufficiently active older people (iii) design activities with both groups to re-design Pacer. Results: Developing physical activity apps for older people should consider the following features: free application, simple interface, motivational messages using audio and visual information, sharing information among users, multimedia input and sharing and user customisation. In particular, we recommend that exercise apps in low-income communities be tailored to our four categories of users differing in baseline physical activity and digital engagement, to match the social and behavioural preferences we discovered.

## Introduction

Population aging is accelerating worldwide. This phenomenon which occurred primarily in developed countries, is now happening in developing countries as well, with a gradual growth

05649-9. L. J. L. received from the funder Coordination for the Improvement of Higher Education Personnel (CAPES) N˚ 001 and from the funder São Paulo Research Foundation (FAPESP) N˚ 2019/02829-9. L. B. C. received from the funder Coordination for the Improvement of Higher Education Personnel (CAPES) N˚ 001 and São Paulo Research Foundation (FAPESP) N˚ 2018/19270-1. P. B. received from the funder Coordination for the Improvement of Higher Education Personnel (CAPES) N˚ 001. V. Z. D. received from the funder São Paulo Research Foundation (FAPESP) N˚ 2016/50249-3. P. C. C. received from the funder São Paulo Research Foundation (FAPESP) N˚ 2016/50249-3. G. A. O. G. received from the funder São Paulo Research Foundation (FAPESP) N˚ 2016/50249-3. The funders didn't play any role in the study design.

**Competing interests:** The authors have declared that no competing interests exist.

in the age groups of older people in the total of the population. This can be explained by the demographic transition, which begins with the decline in mortality and birth rates, accentuated by technological advances in medical sciences and by access to better living conditions [1, 2]. Despite these advances being considered an important achievement for the human legacy, especially for developing countries, such as Brazil (with 13% of aged population) [3], there is still a lack of studies that support social and public health policies that consider a contingent of the population that ages in a situation of social vulnerability [4].

Social vulnerability is defined in the literature as a multidimensional construct, in which behavioral, socio-cultural, economic and political patterns interact with biological processes throughout life [5]. These biological processes mean the accumulation of molecular and cellular damage that lead to the reduction of physiological reserves and increase the risk of chronic diseases (CD), such as cardiovascular and Type II Diabetes Mellitus [6]. These chronic conditions if not treated lead older adults to develop the frailty syndrome, in which "compression of morbidity" is considered one of the greatest public health challenges [7] and is equally onerous to health economics worldwide [8]

The chronic conditions compromise the performance of activities of daily living (ADL), such as feeding, dressing, bathing, sphincter control, sexual activity, self-care and mobility, with significant impacts on functional capacity and well-being in old age, being very prevalent in older adults [9, 10]. However, these functions can be strengthened and rescued through regular and guided physical activity (PA), which has been shown to have a positive impact on the healthy aging process and to decrease the risk of chronic conditions [11]. More specifically, data from studies suggest that physically active individuals increase their chances of living a healthier life at older ages when compared to inactive ones [12].

Despite this, physical activity levels remain insufficient in older adults with low education in Brazil. In 2019, the percentage of Brazilian adults who were sufficiently active in leisure, that is, who practice more than 150 minutes of moderate physical activity or 75 minutes of vigorous physical activity in the week, was only 39.0%. Furthermore, the this practice tends to decrease with age: 49.4% in the range of 18 to 24 years and 24.4% in adults aged 65 and over. It also, which varies with schooling, from 25.8% (8 years of study) to 50.0% (with 12 years or more) [13].

Motivating people for exercise has been one of the biggest obstacles to active life, since the behavior change process is not linear and requires individualized monitoring [14, 15]. Thus, identifying the behavioral state of physical activity can assist in the design of interventions. The Transteoric Model (TTM) for behavior change, for example, identifies five behavioral states of physical activity: (1) Pre-contemplation, who are neither active nor intend to change behavior. (2) Contemplation, individuals who are not physically active, but intend to start an activity. (3) Preparation, when they begin to develop activity slowly and gradually to desirable levels. (4) Action, those who already practice according to the recommended level for less than six months. (5) Maintenance, when the individual has been practising for more than six month [16].

The application of the TTM for sedentary behaviors can provide crucial and in-depth information about participants' readiness to change their sedentary behaviors. A recent approach to persuasion is the use of mobile health applications [17, 18]. Behavior change carried can be triggered and monitoring using smartphones that send motivational messages to an individual in order to shape their attitude or behavior [19]. Unfortunately, 50% of mobile fitness users stop using their fitness-tracking device after six months [20]. This is due to the lack of behavior change constructions that are necessary for long-term engagement [21, 22].

The uptake of mobile technologies in the market, as well as advances in their applications, offers a potential solution to support inactive individuals to improve their activity levels. More

than half of the worldwide population now owns a mobile phone and, in most regions, at least 50% have access to the internet [23]. In Brazil, there are already more active smartphones than people [24]. Despite the proven effectiveness of mHealth programs in supporting behavior change [25], there is still a dearth of research focusing on mHealth for people in vulnerable situations [26, 27]. Thus, it is important that strategies and interventions are adaptable to the needs of this population and designed through a process of co-creation.

Co-design is a technique in which potential end users in partnership with other stakeholders and researchers work together in all stages of product development, from needs assessment, content definition, prototyping, testing and dissemination [28]. The dynamics offered by this technique allow for conceptual re-development and strategies based on the socio-cultural needs of different groups [29, 30]. Relatively new in the area of health, the concept has typically been used in projects to improve service to patients [28]. Nevertheless, it makes sense to consider this process for the development of interventions in mHealth research, especially now with the expansion of access to mobile technologies [31].

The market for this type of applications has spread rapidly; however, the evidence from research on the development and evaluation of these applications is still incipient in populations living in vulnerable situations. The main aim and contribution of the present study is to identify the requirements of older Brazilian people in a low-income community for mobile exercise apps. The co-design process enabled the development of product concepts with and for this population, to improve their physical activity in urban environments.

## Materials and methods

### Design

The study received ethics approval from the Federal University of Sao Carlos Ethics Committee on Human Experimentation (CAAE 81121317.3.0000.5504). Participation was voluntary and complete anonymity of participants was assured. All participants signed the Free and Informed Consent Form and received a copy of it in accordance with the guidelines and regulatory standards for research involving human beings.

This qualitative focus group study enrolled a purposive sample of community-dwelling participants aged between 40 and 90 years old registered in primary health care centres from a region of low income and high social vulnerability in the countryside (Sao Carlos town) of Sao Paulo State in Brazil. Participants were selected to vary in level of physical activity as described below, and divided into two main focus groups of Sufficiently Active and Insufficiently Active. They were later subdivided within each group based on their level of digital engagement. This was to perform a creative co-design activity using the Focusgroup+ method [32]. This is described under Procedures section below, but essentially involves use of a conventional focus group to collect critical feedback on product concepts, before challenging sub-groups to re-design those concepts themselves.

Databases from two previous studies were used in order to randomly select individuals over 40 years old to be included in the study. The studies were a controlled and quasi-experimental research for adults and elderly individuals in the locality of the present study [33]. And a cross-sectional study carried out with elderly people from the same region [34]. Three undergraduate students contacted the individuals who met the eligibility criteria via phone in order to invite them to attend the two focus groups until a total of 24 participants were obtained (12 sufficiently active and 12 insufficiently active). To obtain this quantity in each group, 136 individuals were contacted. Their previous level of physical activity was verified in the database.

The inclusion criteria were: participant being aged between 40 and 90 years; comprehension and verbal communication skills; agree to participate in the research, with the signature

of the Informed Consent; and participant be registered in the local Family Health Strategy. The only criterion for the exclusion of study participants was living outside the indicated context (region).

Of the 136 individuals registered in the database, 87 did not agree to participate, 25 said that they would maybe attend and 24 accepted. The calls were made in January 2018 and 24 confirmed participants were invited via phone call to participate in a pre-interview in their homes. Two of them were not found after three attempts and the 22 who were interviewed were included and then divided into Active and Insufficiently Active, according to The International Physical Activity Questionnaire as follows: ACTIVE, for participants who summed for any added activity: ≥ 5 days / week and ≥ 150 minutes per week (walk + moderate + vigorous) and INSUFFICIENTLY ACTIVE for participants who scored below 150 minutes per week [35]. In the case of 50% of dropouts, we could have groups of at least six people, according to the minimum number advised by the developers of the Focus group+ method [32].

The pre-interviews made in the homes of the participants were carried out by undergraduate students, authors of this paper (Bonadio, Bet, Lorenzi, Paixao). The questionnaire had five sessions: (i) sociodemographic information (such as name, gender, age, education, occupation and address), (ii) items from the International Physical Activity Questionnaire (lPAQ) [35] to evaluate the level of physical activity in Active Sufficiently and Insufficiently Active, (iii) items from the Behavioral Regulation in Exercise Questionnaire (BREQ 3) [36] about the individual perception about the practice of physical exercise, (iv) digital engagement information about (cell phone and smartphone use internet access, frequency of access, applications use and preferences, etc), (v) as well as semi-structured and open questions about the practice of physical activity (these questions are related to the individual having practiced physical activity at some time in their lives, currently practicing or never having practiced). The complete questionnaire is available as additional material.

In a subsequent co-design activity, the participants were sub grouped in digitally engaged and digitally unengaged, according to the frequency of using smartphones and internet access. All participants who used a smartphone more than twice a week were considered engaged. The final sample of individuals who were present in the focus group was composed of 17 participants (5 no-shows) who responded to the pre-interview and participated in all stages of the focus groups. The characteristics and subgroup of participants in the final sample are described in Table 1.

**Table 1. Sociodemographic characteristics, digital engagement and level of physical activity of the participants in the focus groups.**

|  |  | Active | Insufficiently Active | Total |
|---|---|---|---|---|
| Age(years) | Median | 63 | 66 | 63 |
|  | Min | 44 | 52 | 44 |
|  | Max | 73 | 88 | 88 |
| Gender | Female | 8 | 4 | 12 |
|  | Male | 4 | 1 | 5 |
| School years | Median | 4 | 4 | 4 |
|  | Min | 0 | 3 | 0 |
|  | Max | 8 | 8 | 8 |
| Digital Engagement | Digitally Engaged | 6 | 2 | 8 |
|  | Digitally Unegaged | 6 | 3 | 9 |
| Physical Activity Level | Media (±SD) | 335.9 | 42.0 |  |
| Minutes per week (n = 16) |  | (±192.7) | (±36.5) |  |

São Carlos, São Paulo, Brazil. 2018.

## Procedures

Two Focusgroup+ workshops were carried out in the following three phases: (1) general discussion of motivation to exercise following a short video, (2) introduction of a mobile exercise app called PACER ® for critical feedback, and (3) re-design of PACER by two subgroups. PACER was selected as one of the most popular exercise apps on the Google Play Store having pedometer and coach functions. All of these steps were performed on the same day, in a single workshop session lasting about 3 hours, with a short break between each phase. Sessions were held in an event hall of a well-known church in the region where the participants lived. They were facilitated by two PhD professors (Castro and Gomes) with help from Masters and Undergraduate students (Bonadio, Bet, Lorenzi, Paixao).

At the beginning of phase 1, the researchers first explained the Playful Datadriven Active Urban Living (PAUL) project to the participants, as well as the co-design methodology. The PAUL project is a partnership between the Federal University of Sao Carlos (UFSCar), Federal University of Sao Paulo (UNIFESP), and the University of Amsterdam. This project has the main purpose to verify the effectiveness of an application for physical activity that has elements of gamefication and messages, and to change behavior by increasing the practice of physical activity. Active and healthy aging, motivation for physical activities in urban areas and applications for monitoring and training were also addressed and discussed during the workshop. All participants were non-users of physical activity applications.

After the project description, participants were played a short video by a celebrity Brazilian doctor, in which he described his positive experience of taking up running at the age of 50. This strategy was thought to be a good introduction to the discussion, raising issues about benefits and practicalities of physical activity in the urban environment. Subsequently, one of the researchers asked a first set of questions about:

1. What do you think about having a coach to help with the exercise?

2. Do you think motivational messages would help your practice? Why?

3. Some people like to play against themselves and improve their own performance, some like to play competitively against others and some like to play as a team. What practical type do you prefer? Can you describe the way you like to play and measure your performance? Would like to share your achievements / activities in social media?

4. Do you think that an activity goal will motivate you to accomplish more? What would be the requirements of a good activity goal? Do you prefer a self-defined goal or an automatic (adapted) goal?

5. Virtual Rewards vs. real of life, motivate? Amount? Points in a game / message congratulating you on the achievement?

In the second and third phases of the workshop, we introduced a commercially available exercise app called PACER Pedometer and Coach to the group for feedback and re-design. This technique was developed as an accessible way of doing co-design with a group of older adults who may be unfamiliar with new technology. Participants are shown a product concept for verbal feedback in the usual way. They are then split into subgroups and asked to consider what they would like to keep, lose and change about an existing product or product concept, as a precursor to deciding features of a new concept that they prefer [32]. The subgroups have to reach consensus within a fixed time, aided by a facilitator. In this study, participants were split into smaller groups of about five people and given one hour to come up with a new design. The designs of each sub-group were then shared with the wider group for further discussion.

In phase 2, PACER was shown to participants by projection, indicating step-by-step how to use some specific tools of Pacer, to explore the content guided by the researchers. The app was then made available on a number of Android tablets for participants to manipulate themselves. The basic functionality of a Tablet was initially presented, so that subjects could use the tool with maximum autonomy. Individuals performed some tasks and then gave their opinion before building on the new design according to their preferences, improving and changing what they had seen. The app is from the health and fitness selection, described in the PlayStore as pedometer, health, weight loss and fitness. It has tools such as goal setting, device synchronization, physical conditioning device, training monitoring, among others.

The intention was to show the variety of resources within these kind of applications for Physical Activity, for subjects to select from according to their preferences in the process of co-creating one or more new product concepts. These design concepts would then form the basis of a professional design process to create a prototype app combining the best features and ideas in partnership with colleagues in the Netherlands.

After the demonstration, and free use of PACER by participants another round of questions was asked to obtain initial reactions and critical feedback:

1. What do you think of this? Would you use? Is it useless? Why?

2. Would an application for physical activity be effective in stimulating practice?

3. Would you like the active participation of your application data?

4. Would you like to talk to others who also do activities with you within the application?

Participants then expressed their opinion, answered the questions and discussed with each other.

After a break, the phase 3 re-design activity was then started, in which the participants considered what to Keep, Lose or Change about the app. This led to further design ideas for a new version of the app which was preferred by the whole group. Design ideas were collected through drawings and post it notes, where the participants drew and wrote down their ideas. They also made comments on themes, and ideas for tools which were video recorded. Finally, each sub-group showed their new product concepts to each other, and discussed them further. This procedure was repeated for both the Sufficiently Active and Insufficiently Active groups, resulting in four re-designed versions of PACER across the four subgroups.

## Data analysis

Descriptive analyses, such as median, minimum and maximum values, were performed to characterise the sample in age, gender, education, physical activity level, and digital engagement. The comparisons between the groups in the analysis were organised in order to: Compare the ACTIVES group in general, looking at their subgroups of engaged and not digitally engaged, also the INSUFFICIENTLY ACTIVE in the same modality. When then compared the subgroups by equal pairs, comparing active and insufficiently active with engaged-engaged and non-engaged-non-engaged. Comparisons for active-insufficiently active groups such as large groups were also designed.

The organisation of the Content Analysis was carried out according to [37]. A set of communication analysis techniques was performed aiming to obtain, by systematic and objective procedures for describing the content of messages, indicators that allow inference of knowledge regarding production/reception conditions (inferred variables) of these messages points. This analysis was divided into (1) pre-analysis; (2) exploration of the material; (3) data processing and interpretation.

**Pre-analysis.** In this stage, the study was based on the transcription of the recordings made in the intervention, the speeches of the researchers and research subjects. The transcription was organised by groups (level of physical activity and digital engagement), enabling comparisons. The corpus was built, a set of relevant documents that were submitted to the analytical processes. Subsequently, the hypotheses and objectives were formulated. However, the present study was carried out "blindly", without preconceived ideas. Afterwards, the referencing of the indexes and the elaboration of the indicators were elaborated. In this, the texts are considered as a manifestation, containing indexes that the analysis pointed out, for example, it may be an explicit mention to a digital resource that influences/motivates the practice, one of the objectives of the study. Finally, the material was prepared. Formal preparation and editing were part of the process, such as (a) interviews recorded and transcribed in full; (b) material resources such as evaluations, response rates organised and catalogued. The editing of texts went through computer processing, prepared and coded.

**Exploration of the material.** After consistent ordering in the pre-analysis, the next aspect was the systematic administration of the data, coding operations, enumeration.

**Data processing and interpretation.** Frequency analysis was also carried out, which accounted for the repetition of content common to most respondents. The interpretation used thematic analysis as a tool, counting one or several themes/items of significance, creating comparisons and elements of customisation based on the aspects brought by the individuals. The categories and themes from the transcription and were analysed by two of the authors (Castro and Frohlich). As for the division, the analysis was carried out with the database divided into: (a) Five questions exposed for debate, about: motivational video for physical activity practice in an urban environment spent at the beginning of the intervention; also about the practice of physical activity and the elements that motivate people to practice (for example, goals, objectives, incentive, coach, monitoring); (b) Four questions about the physical activity application were used as a basis for re-design PACER $^{®}$ about how this digital tool can influence the elderly to practice physical activities, whether the resources available in the app contribute to motivation or not. This topic corresponds to the second part. The main objective of this division is to understand the significance attributed to the responses of the subjects in two data collections: (a) Aspects that motivate the practice of physical activity for these individuals, not necessarily digital, such as the environment, social interaction, trainer, objectives, goals, groups, forms of incentive; (b) Necessarily digital aspects, on the issues raised by the subjects who sought to complete references of aptitude, or not, of the Pacer$^{®}$ application in motivating the practice.

From the re-design activity, we analysed the values of each subgroups as reflected by their choice of what to keep, lose or change about the PACER app. We also tried to characterize their final new concept with a name and illustration based on the discussion.

## Results

The final sample of the study was composed by 17 participants, being divided into two groups: physically active (12 participants) and insufficiently active (five participants). Both groups were subdivided and classified according to the participants' digital engagement. The first group was composed of six people equally in the subdivision, while in the second, two were considered digitally engaged and three were not engaged. The main results found in this study point out that the development of physical activity applications for older users consider the following resources: free application, simple interface, motivational messages using audio and pictures, information exchange between users, multimedia input and sharing and user customisation. These insights can be important for designing intervention applications to improve

**Table 2. Frequency analysis of focus groups questions.**

| Category | Active | | Ins. Active | |
|---|---|---|---|---|
| | Yes | No | Yes | No |
| Trainer to help in performing | 4 | 1 | 4 | 0 |
| Motivational messages | 5 | 0 | 4 | 0 |
| Collaboration | 3 | 0 | 4 | 0 |
| Competition | 2 | 0 | 0 | 4 |
| Loners | 3 | 0 | 0 | 0 |
| Like to share on social media | 8 | 0 | 5 | 0 |
| Set goals | 8 | 0 | 2 | 0 |
| Prizes | 5 | 0 | 4 | 0 |
| Ranking | 2 | 1 | 0 | 0 |

healthy behaviour, considering a universal product, but still adapted, increasing the likelihood of adoption.

Table 2 summarises the results of the frequency analysis of focus group questions, showing the repetition of content common to most respondents. Note that the functions that participants would most like to have in the application were "like to share on social media" and "set goals".

Below, we present and discuss the general results of the study, divided into the following categories: Focus Group and Re-design.

## Focus group

**Motivation to exercise.** The first question referred to the **importance of a personal trainer or guidance to assist in the physical exercise performance**. Most people in the sufficiently active group indicated the trainer as a motivating factor for the practice of physical activities. However, one participant commented that the financial reality of people often does not allow access to this type of service. The insufficiently active group also agreed on the importance of having a coach. During the discussion, participants pointed out that the absence of a teacher to assist during physical activity may lead to the abandonment of the practice. In addition, they commented on the importance of a person who can customise the program considering frequency and intensity of the exercises due to the diversity of older individuals. Besides motivational purposes, the presence of a personal trainer has a safety dimension. Most participants are afraid of hurting themselves during physical activity without proper guidance, especially the beginners.

*"One person (coach) should monitors the hasty walkers and another those slower ones. . ., so when I started walking I gave up because of it, whenever my blood pressure went up I felt sick and my group would go and leave me behind"*

—Insufficiently Active Participant.

*"At least at the beginning you've got to have someone to check on you, so you won't go exercising wrong, because sometimes you might try exercising to heal something and you end up injuring something else."*

—Insufficiently Active Participant.

The next question concerned the **effectiveness of motivational messages during exercise**. In the group of sufficiently active, the importance of "face to face" feedback was highlighted. In this respect, the participants preferred that a person or an avatar encourage them with praise. On the other hand, the written messages were not so popular. When questioned about text or audio messages, the vast majority signalled preference for audio messages, because these were expected from a coach avatar and for literacy reasons, since some participants did not know how to read, and most have very few school years, as shown in Table 1. In the group insufficiently active, the majority reported that motivational messages help, which is an important incentive to praise the activity. About timing, they prefer to get the messages when finishing the activity, so they did not interfere with their rhythm. They also reported that text and audio messages encouraged, but there was also a preference for audio messages in that group.

*"I do think so because communication is good for feeling accomplished, seeing a smile, listening to a "let's go" or "this will be good for your health".*

*"If it won't get in the way, cause sometimes you might be at your peak on a walking and you won't stop to check your phone"*

—Insufficiently Active Participants.

Considering **modalities of competition, collaboration or solo activities**, the sufficiently active group pointed out a preference for collaboration, i.e. one helping the other, but also valued solo activities. Participants said that other people may make them unfocused or talkative, which can slow down the rhythm of activities. The group of the insufficiently active reported the same issue of rhythm. They even commented that two groups with a coach are important: one for younger and / or faster people and another for people who cannot keep up.

*"You wanna walk, right? So you got your limit, having someone else would get you off the moment. When you talk, you won't catch up to the rhythm you need to. So having a person beside you that won't keep up with you makes you stuck with someone else's rhythm"*

—Active Participant.

*"I think it should be two groups, girl, one for that young ladies like you that won't wait for me, breathless, no, then I'd be ruining their walking, because they're young and active, willing to go faster, got it? I would want to go slower, got me? . . .Oh, now talking about me, when I started I saw the girls excited about it and decided to give it a try"*

—Insufficiently Active Participant.

The next step of the debate was to find out if the **establishment of goals and marks motivates the practice of physical activity**. Both groups agreed that it is of great importance, since it is possible to account for the objectives achieved.

*"It's good for people's health, right?"*

—Active Participant.

The next topic was more debated than the previous one and concerns the **actual and virtual rewards when a goal/achievement is achieved**. Members of the active group liked it

especially when the researcher suggested the possibility of financial rewards: *"Then I'd do many miles, yes \*laughter\*"*.

As for other rewards, such as praise, images and virtual prizes, participants also pointed out that they encourage them by recognizing what has been achieved. Being recognized when you achieve a conquest proved to be essential. The Insufficiently Active group also noted the importance of rewards. They like financial rewards, but also virtual rewards. What differs from each other in this dimension is the importance of the positive psychological effect that the rewards cause. For those, what matters is the feeling of being rewarded, not the product of the prize itself.

> *"A trophy, I guess, or any other present one might get, no matter the size of it, any little thing is worthy, that's what matters to us, even if it's just a reminder. It pushes us through, like I said, let's say you go walking, for everything they've done so far in this activity, it must be valued, right?"*

—Active participant.

> *"I feel just fine about my health, my mind"*

—Insufficiently Active Participants.

Other interesting aspects that were not part of the issues described in the method emerged from the debate and were highly relevant to the results. One is about the **difference in physical activity practice in urban versus rural settings**. The group of sufficiently active raised this topic. For them it is easier to carry out the activities in rural areas due to the open spaces. In cities, according to these participants, there are not many places that are inviting to practice, such as green squares/campuses/gyms. In rural areas green and open spaces are more available.

> *"I saw a man in his late 90s, I think he's 97, who goes walking every single day. He used to work in farms, from sunrise to sunset, he got strong doing so, sometimes farming is the only exercise one has access to"*

—Active Participant.

> *"It's quieter, I couldn't exercise in the city, where I used to live there was nowhere to go, it didn't by the time I got here, but I wouldn't go out, back then I used to go walking, I'd go out and back, I'd go to my mom's, but there is no place to go".*

> *"Nowhere! There was nowhere to go! While in the country there were space, there was work, the fields, the cricket by the beach, I was always walking back and forth at the fields by the river"*

—Active Participants.

Taking advantage of this discussion, the researcher questioned how it would be a tempting space to the practice of physical activity. In response, they pointed out that a gym with rooms and a coach to guide the practice would be the most appropriate.

> *"There are the rooms, right? There's a good space for people, the lady helps us just as at the gym"*

—Active Participant.

Another aspect of great importance that both groups brought to debate, both active and insufficiently active, was **public security**. The women reported fear of walking alone, due to robberies, harassment and traffic violence (not accessible sidewalks).

*"I used to go walking everyday by 5:30am, a one-hour walking, I'd go from my place to the track, but some things started happening, you know, some men were chasing the ladies there, so I stopped going because I was afraid, because there was only me and another lady with me. But I think, like, to go walking in two or three is good, because we go talking and distracting each other, we open up, and that was good for me. But I stopped because of that and I would go back, I'd want it if it were safe".*

—Active participants.

*"If you go by yourself, whichever street you go isn't cool (safe)"*

—Insufficiently Active Participant.

At the end of the first part of the intervention, music during physical activity practice was raised by the active group as a motivation factor: *"Music cheers the activity up, right, it pushes the rhythm"*—Active Participant

The main findings were that most participants would like to have a personal trainer to help them performing physical activity even if that trainer is an artificial intelligence with demos and audio messages for guidance. All of them would like to receive motivational messages. Some wanted these during exercising while others wanted them as a reminder to start the activity. They thought it would be interesting to received real life prizes, but more importantly to get messages of praise and/or virtual prizes. They valued the possibility of setting personal goals and monitoring trends and progress for themselves (the loners), to share on social media (the collaborators gamers) or to post on rankings (the competitors).

**App feedback.**  The second part of the study was about the feedback questions of a commercially available application (PACER), used as a basis for re-design. In order to compare the findings, the subgroups were separated by digital engagement, as follows.

(1) Sufficiently active group—digitally engaged and unengaged

After the demonstration about the Pacer application, the researcher questioned the participants' opinion, specially about usefulness. In the group of digitally engaged, some people answered affirmatively, they found it interesting, however without further discussion. In the non-engaged group there were also affirmative answers, however, some interviewees were not familiar with the device and expected it to be difficult to use.

The next aspect debated was the effectiveness of the components to stimulate the practice. Those digitally engaged liked goals such as weight loss and walking tracking. When questioned about the stimulus to the practice, participants pointed out that they would feel safer if the information came from a health care professional, even agreed to an avatar within the application being a doctor or other health professional.

The unengaged group also valued the same goals (weight loss and walking), some commented that they like to perform their activities with outdoor gym at the park. We therefore asked participants' opinions about communicating with other people who also do activities within the application. The group of digitally engaged people said that sharing and talking to one group is an added incentive. They also asked whether the application can communicate the network with the places that individuals are walking, doing geographic tracking.

*"Is there a way to communicate with the place we walk on?"*

—Active and Engaged Participant.

In the group of non-digitally engaged, communication with space also appeared, even indirectly.

*"I like exercising at outdoor gyms at the park, with their equipment, sharing encourages people using them"*

—Active Unengaged Participant.

In general, individuals from both groups liked to chat within the app, share photos, and also communicate with the environment.

(2) Insufficiently active group—digitally engaged and unengaged

The first question was "What do you think of this? Would you use? Is this helpful? Why?" The engaged group has partially agreed that it is useful. In the unengaged sub-group only one person said that it is useful, the others reported having difficulty in using the phone, so they could not comment.

*"I wouldn't add or remove a single thing because I don't know how to work them out, but I'm very willing to learn"*

—Active Unengaged Participant.

As for the aspects seen as effective to stimulate the practice, there was great surprise in the group of the digitally engaged because people did not know the resources of an application for physical activity, however, they were happy with the new discovery.

*"I think it's very interesting. I've never heard of an app that shows calories. . . That's right? Burnt calories, the number of steps taken, it tracks our progress, as you said, look how different!"*

—Active Unengaged Participant.

For this unengaged group, the most meaningful function is the guidance for exercising. Other valued aspect that the application offers is the encouraging, especially in setting the goals, because it is possible to choose them.

*"I like it because it teaches us step by step, so we get to learn"*

—Active Unengaged Participant.

When asked about talking to other people within the application, the group of digitally engaged people pointed out that can be very useful to share photos of healthy eating to encourage changes to healthy habits in others and themselves.

*"I think sharing is very good, I always share photos of my lunch with my sister and say "here, sister, my salad plate" because she likes fried food and stuff like that, so we try to push her into eating more salad and stuff."*

— Active engaged Participant.

This group also showed interest in connecting the physical activity application with social media to share with friends.

*"Does it connect with Facebook as you told us? Can you show how to do it?".*

—Active engaged Participant.

On the other hand, the digitally unengaged group emphasized only the importance of talking and interacting with others.

*"Yes, interaction is what matters".*

—Active Unengaged Participant.

In addition, comparing two groups, a question dealt with the effective aspects to stimulate practice. The active group scored on the goals of weight loss and the walking modality, while the insufficiently active ones scored on the calorie marking, amount of steps, weight and the follow-up of the progress weekly by the application. They also rated the motivational messages as a positive aspect.

In the issue of communication with other physical activity practitioners within the application, the group of engaged activists responded that this interaction would be motivating, and even reported that it is interesting for the application to communicate with the network of places/environments that individuals are struggling to do side dish. The group of the insufficiently active engaged also responded that this tool is useful, emphasising that it would be in their interest to share the Pacer application, goals and objectives within social media, such as Facebook, to encourage friends.

Another aspect addressed was teaching and learning, participants appreciated the fact that the application teaches then to carry out the activities step by step. Finally, on the issue of talking to other people within the application, both groups said they wanted to share pictures with a group.

## Re-design

**Focus group: Active, digitally engaged.**   This group of participants took part in regular exercise and were also regular users of the internet. Consequently, they began to re-design Pacer as a performance enhancing tool which was easier to wear and more informative during the process of exercise itself. They chose to keep the monitoring of steps and the way Pacer supported the sharing of data within exercise groups. They wanted to ´lose´ the need to pay for the system and felt it should be free to use. The main changes they wanted to introduce included more multimedia input and output in the form voice and photographs, additional automatic sensing of biodata, presentation of messages through a personal trainer avatar and connectivity with beacons in the environment giving environmental information. They also wanted to change the device used when exercising to a wearable device, as then knew how difficult it was to carry a smartphone while exercising. These choices led them to design a "Doutor Sabido" concept, meaning literally doctor knowledge (Fig 1).

The concept of "Doutor Sabido" is a wearable device that continuously monitors outdoor workouts and delivers voice-based information about the environment and the user's performance. After exercising, it can present summary statistics and feedback through dialogue with the avatar coach, and share this information with others in a selected exercise group. The main purpose of the device is to track and reflect back performance information for users to improve.

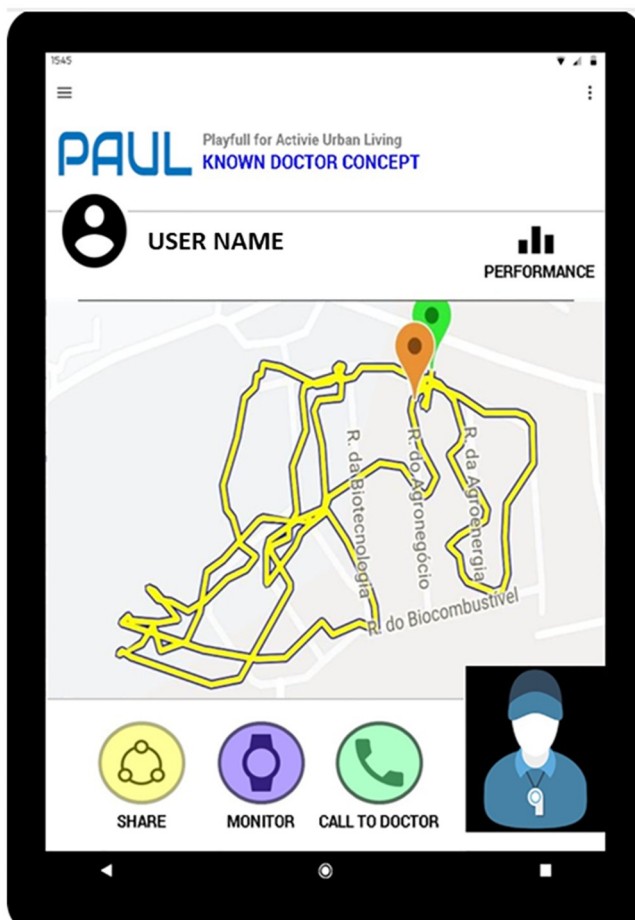

**Fig 1. "Doutor Sabido" concept (doctor knowledge).**

**Focus group: Active, digitally unengaged.** This group of participants took regular exercise but were relatively low users of internet technology. They were most concerned about being able to use an exercise app and whether it would be compatible with the kind of low-end phones and technologies they used. This was reflected in their choices about what to keep, lose and change about the Pacer app. They wanted to keep the step monitoring functionality, messages about goals, food and nutrition, and example exercises. They also liked the fact the app was available to use at any time of the week, including weekends and evenings. However they wanted to lose some complexity in the interface to the app, the need to carry a mobile phone when exercising, the cost of being constantly connected to the internet and having to upgrade the app itself. The main changes they discussed were to improve its compatibility with low end phones, to introduce a personal trainer avatar, and to support voice and photo input and exercise demonstrations. This resulted in a concept called "Grupo Unido", meaning united group (Fig 2).

The "Grupo Unido" concept was designed for a group of friends who can physically exercise together but also share their performance data with others outside the group. The main interface is through a wearable device, which collects step and other data and allows the user to take photographs during their physical activity. The photos might document the activity or the pleasure of doing it with others, and photos can be shared later via a mobile phone along with

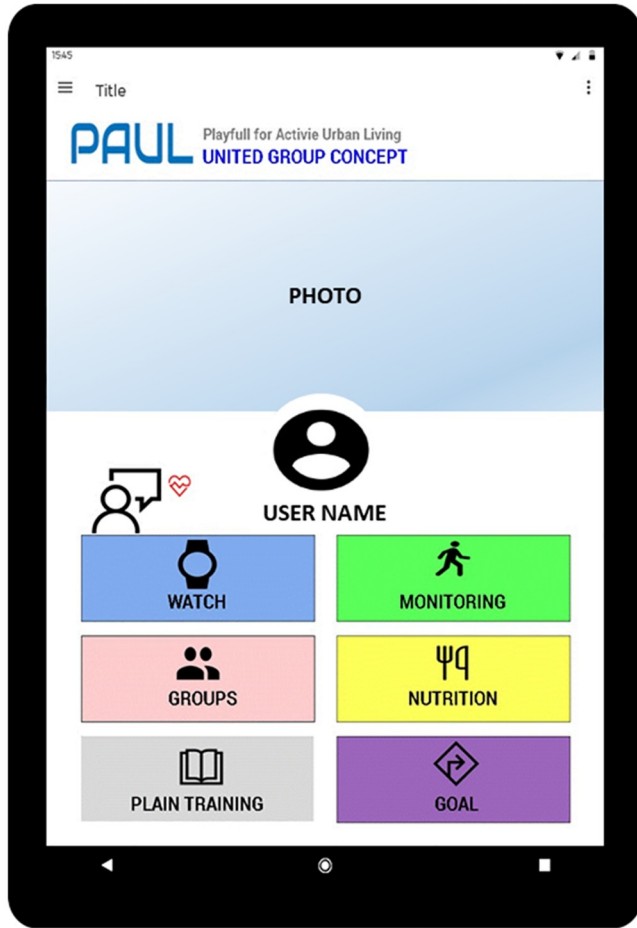

**Fig 2. "Grupo Unido" concept (united group).**

performance data. The wearable device might also have a screen for viewing the face of an avatar coach and hearing motivational voice messages from the coach. The device would be self-contained without internet connectivity to reduce cost and complexity of use.

**Focus group: Insufficiently active, digitally engaged.** Focus group was comprised of less activity participants differing in their levels of digital engagement. This sub-group were frequent users of technology and familiar with the latest smartphones. Because they were relatively inactive, their preoccupation was with motivation to exercise and this was reflected in their choices of what to keep, lose and change. They wanted to keep step monitoring and the collection of other exercise data, and liked the mobile phone interface and device with which they were familiar. However, they wanted to lose the need for payment, and introduce some changes to the design. These included the automatic sensing of a wider variety of data during exercise, more motivational messages including praise for achieving goals, with voice and picture output. This all-female group wanted an attractive male avatar coach to encourage and praise them, and mechanisms for coordinating co-present exercise sessions amongst a wider online group. They also wanted to share progress with others online. This resulted in the "Amigos Unidos" concept, meaning united friends (Fig 3).

"Amigos Unidos" is a mobile phone app designed to support the motivation of exercise first and foremost. Motivation comes from two channels. First, it comes from a supportive avatar

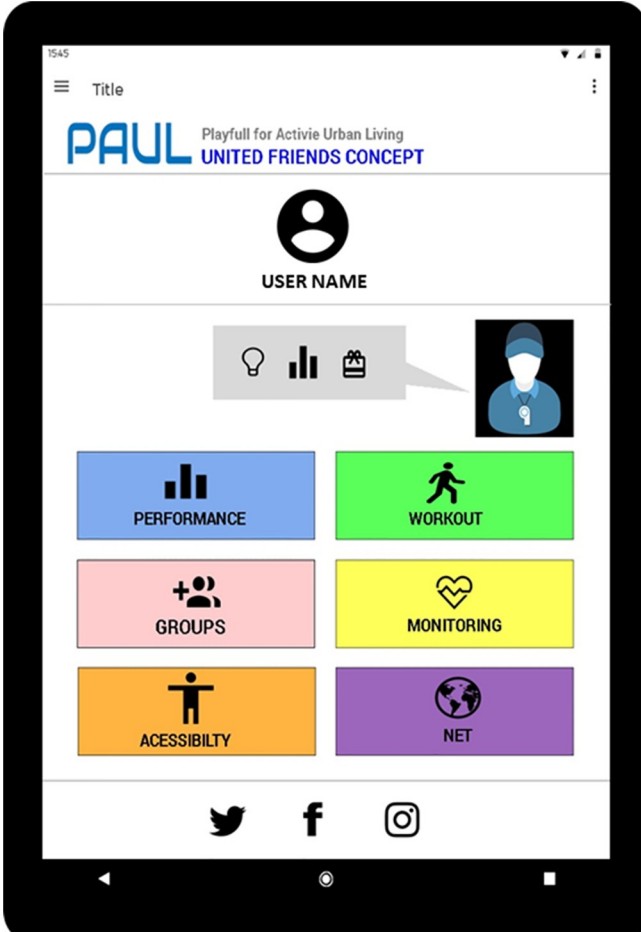

**Fig 3. "Amigos Unidos" concept (united friends).**

coach who can set realistic personalised goals, and then provide encouragement and praise to achieve them. Second, it comes from the user's wider social network who may also be exercising with the app. This is achieved by full integration of "Amigos Unidos" with existing social networking systems such as Facebook and Twitter.

**Focus group: Insufficiently active, digitally unengaged.** This group were relatively inactive and not very engaged with digital technology. They shared a desire to be motivated to exercise with the previous group but wanted a simpler mobile phone app, which could be used outdoors. The wanted to keep step monitoring and a key functionality, performance goals, and the accessible language they thought that Pacer already had. They would also keep the notion of groups for data sharing and especially the graphs used for giving performance feedback. They would lose the need for payment and introduce changes such as the addition of a personal trainer avatar. They has a particular interest in personal safety when exercising and wanted video and photo demonstrations to see how best to move and prevent injury. This resulted in the "Atividade para saúde" concept, meaning health activity (Fig 4).

The "Atividade para Saude" mobile phone app has a simple interface dominated by a personal trainer avatar who is able to give motivational message and performance feedback, but also to illustrate exercises using video and photo demonstrations. It is designed to be used outside with some attention to navigation and the outdoor environment, and later ability to share

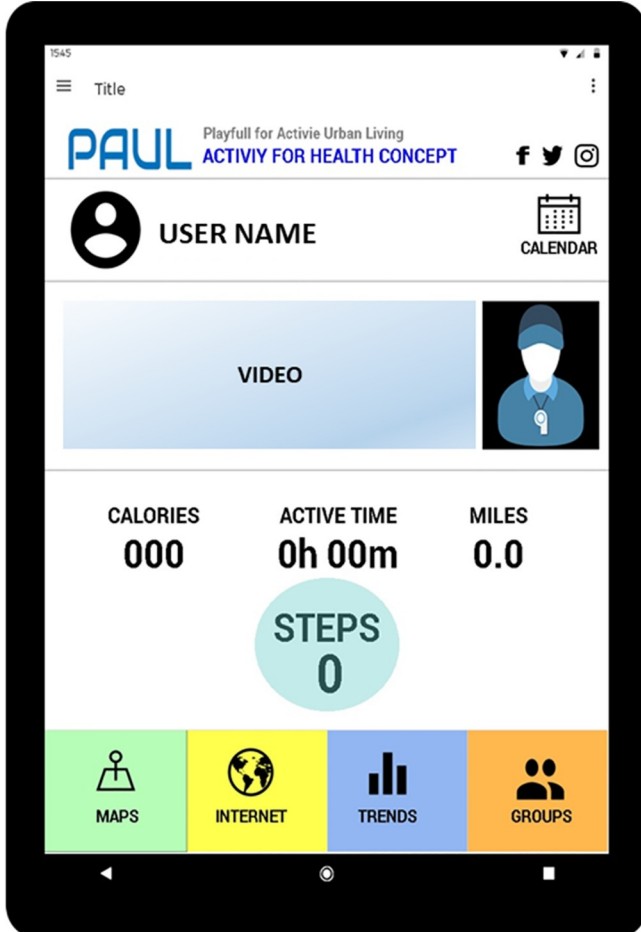

**Fig 4. "Atividade para Saúde" concept (health activity).**

data with others in the users' social network. It can also give graphical feedback on performance to help users understand their exercise patterns and progress.

## Discussion

The present investigation aimed to identify the components necessary for the customization of an application to encourage physical activity, based on the wishes and experiences of users over 40 years of age living in a community of high social vulnerability in Brazil. The main results showed that insufficiently active and active people require different approaches to creation of a physical activity app, but within the same app framework. The use of mobile technology in physical activity research is still incipient. However the enhancement of physical activity apps supporting a variety of behaviour changes strategies is very relevant, and more likely to produce long-term improvements in physical activity level [21, 38].

This study used focus group and re-design methods [39] modelling the need for formative evaluation in health-related smartphone apps. Focus groups have traditionally been used in market and design research [40]. This is a good strategy here since such groups can be recruited to contain the main users and their different characteristics. Adding re-design methods following focus groups are especially empowering for digitally unengaged users, whose

voice and creativity might otherwise be ignored by prioritizing digitally engaged 'early adopters' of technology [32].

During the co-design methods, the digitally unengaged users indicated a simpler interface. All groups would like to personalize the set-up and operation of a single app. The co-design method is likely to improve usability because users can give their opinion about their own desired use [39].

Independently of the physical activity level, all participants wanted free apps. This may have happened because the sample was specifically from a low income community. [41] showed that presence of behavior changes techniques varied by app type and price, however studies including interventions with the paid app are more effective than non-paid app.

This study shows that the guidelines for usability and accessibility to illiterate and older people require careful attention. Their comments and suggestions show a preference for messages using sound and image rather than text description. Older women, low income and low education individuals are most likely to be physically inactive. Since changing their behavior is more challenging, the app should use different health technologies to be more powerful [42].

Independently of being active or digitally engaged, all participants consider motivational messages as an essential strategy, which are strongly recommended in most behaviour changes theories [16, 38, 43, 44]. The greatest advantage of technology is the fast feedback of information. Dynamic feedback system theories of health behaviour can be developed using longitudinal data from mobiles devices and control systems engineering models [45]. This fact can be a great differentiator between monitoring using traditional methods and mobile devices.

Our main goal was to show how people over 40 years from low income communities think about usability of a physical activity app, and begin to specify its interface and functionality for themselves. Future work might use these specifications in the design, implementation and evaluation of better apps for physical exercise by this population. The dynamic approach adopted in this study allowed the participation of people with different profiles, which made the results more plural and more likely to be assertive. In addition, the facilitator's role was important in providing support and ensuring the involvement of all those present. Still, the methodology developed enabled the identification of several factors that must be considered in the process of creating an application to encourage physical activity. In this way, this study brings with it key elements for creating applications focused on the needs and desires of users, and innovates by bringing new findings to the scientific community, by conducting a study with a robust methodology, in a region of high vulnerability with participants over 40 years old and with a low level of education.

The limitations of the study are that it enrolled only people over 40 years old in regions of high social vulnerability, so there was no inclusion of different individuals to compare with this population and check if there are different needs and specifications across other age categories or incomes. In addition, engagement and participation in the focus group was higher among active compare to the non-active group, it probably happened because the first group was more open to discussing physical activity itself. Other studies should find ways to encourage non-active people to engage in the debate and participate in the design of the app.

Creation of physical activity apps to low income community should consider the following features: free app, simple interface, motivational messages using sound and image, changing of information among users and customization of the app by the user. These characteristics might increase usage and maintenance of health behavior change in the long term. These findings serve as an important foundation for informing the development of appropriate and efficient intervention techniques using such technology to enhance this important health behavior.

## Supporting information

**S1 File.**
(PDF)

**S2 File.**
(PDF)

**S1 Questionnaire.**
(PDF)

**S2 Questionnaire.**
(PDF)

**S1 Transcript.**
(PDF)

**S2 Transcript.**
(PDF)

## Author Contributions

**Conceptualization:** Paula Costa Castro, Lua Bonadio Romano, David Frohlich, Lorena Jorge Lorenzi, Lucas Bueno Campos, Andresa Paixão, Marije Deutekom, Ben Krose, Victor Zuniga Dourado.

**Data curation:** Lua Bonadio Romano, Lorena Jorge Lorenzi, Andresa Paixão, Patrícia Bet, Grace Angélica de Oliveira Gomes.

**Formal analysis:** Paula Costa Castro, Lua Bonadio Romano, Lorena Jorge Lorenzi, Andresa Paixão, Patrícia Bet, Grace Angélica de Oliveira Gomes.

**Funding acquisition:** Paula Costa Castro, Lua Bonadio Romano, Lorena Jorge Lorenzi, Lucas Bueno Campos, Victor Zuniga Dourado, Grace Angélica de Oliveira Gomes.

**Investigation:** Lua Bonadio Romano, David Frohlich, Lorena Jorge Lorenzi, Lucas Bueno Campos, Marije Deutekom, Ben Krose, Grace Angélica de Oliveira Gomes.

**Methodology:** Paula Costa Castro, Lua Bonadio Romano, David Frohlich, Lorena Jorge Lorenzi, Lucas Bueno Campos, Patrícia Bet, Marije Deutekom, Ben Krose, Victor Zuniga Dourado, Grace Angélica de Oliveira Gomes.

**Project administration:** Paula Costa Castro, Victor Zuniga Dourado, Grace Angélica de Oliveira Gomes.

**Supervision:** Paula Costa Castro.

**Visualization:** David Frohlich, Lorena Jorge Lorenzi, Lucas Bueno Campos, Andresa Paixão, Patrícia Bet, Marije Deutekom, Ben Krose, Victor Zuniga Dourado, Grace Angélica de Oliveira Gomes.

**Writing – original draft:** Paula Costa Castro, Lua Bonadio Romano, David Frohlich, Lorena Jorge Lorenzi, Lucas Bueno Campos, Andresa Paixão, Patrícia Bet, Marije Deutekom, Ben Krose, Victor Zuniga Dourado, Grace Angélica de Oliveira Gomes.

**Writing – review & editing:** David Frohlich, Lorena Jorge Lorenzi, Patrícia Bet.

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
