## [Decision Letter · Decision Letter 0]

27 Jul 2020

PONE-D-20-15049

Tailoring Digital Apps to Support Active Ageing in a Low Income Community

PLOS ONE

Dear Dr. Lorenzi,

Thank you for submitting your manuscript to PLOS ONE. After careful consideration, we feel that it has merit but does not fully meet PLOS ONE’s publication criteria as it currently stands. Therefore, we invite you to submit a revised version of the manuscript that addresses the points raised during the review process.

We look forward to receiving your revised manuscript.

Kind regards,

Jeffrey Jutai

Academic Editor

PLOS ONE

Journal Requirements:

2.We suggest you thoroughly copyedit your manuscript for language usage, spelling, and grammar. If you do not know anyone who can help you do this, you may wish to consider employing a professional scientific editing service.  

3.We note that [Figure(s) 1] in your submission contain [map/satellite] images which may be copyrighted. All PLOS content is published under the Creative Commons Attribution License (CC BY 4.0), which means that the manuscript, images, and Supporting Information files will be freely available online, and any third party is permitted to access, download, copy, distribute, and use these materials in any way, even commercially, with proper attribution. For these reasons, we cannot publish previously copyrighted maps or satellite images created using proprietary data, such as Google software (Google Maps, Street View, and Earth). For more information, see our copyright guidelines: http://journals.plos.org/plosone/s/licenses-and-copyright.

1.    You may seek permission from the original copyright holder of Figure(s) [1] to publish the content specifically under the CC BY 4.0 license. 

4.We note that Figure(s) [2 and 4] in your submission contain copyrighted images. All PLOS content is published under the Creative Commons Attribution License (CC BY 4.0), which means that the manuscript, images, and Supporting Information files will be freely available online, and any third party is permitted to access, download, copy, distribute, and use these materials in any way, even commercially, with proper attribution. For more information, see our copyright guidelines: http://journals.plos.org/plosone/s/licenses-and-copyright.

1.    You may seek permission from the original copyright holder of Figure(s) [2 and 4] to publish the content specifically under the CC BY 4.0 license.

5. Thank you for sending us the data set underlying the results presented in your PLOS ONE submission. We notice that some of the information included in the data set may be potentially identifying. Please ensure that the data shared are in accordance with participant consent and provide only the data that are used in this specific study. To ensure patient confidentiality, we would recommend removing the images containing names. Additional guidance on preparing raw clinical data for publication can be found in our Data Policy FAQs (https://journals.plos.org/plosone/s/data-availability#loc-clinical-data).

6. Please include additional information regarding the survey or questionnaire used in the study and ensure that you have provided sufficient details that others could replicate the analyses. For instance, if you developed a questionnaire as part of this study and it is not under a copyright more restrictive than CC-BY, please include a copy, in both the original language and English, as Supporting Information.

7. Please clarify whether the study was specifically approved by the IRB, and clarify how participants gave consent.

Additional Editor Comments (if provided):

Your manuscript has been recommended for publication pending minor revisions. Please address carefully the reviewers' comments.

Reviewers' comments:

Reviewer's Responses to Questions

**Comments to the Author**

1. Is the manuscript technically sound, and do the data support the conclusions?

Reviewer #1: Partly

2. Has the statistical analysis been performed appropriately and rigorously? 

Reviewer #1: No

3. Have the authors made all data underlying the findings in their manuscript fully available?

Reviewer #1: Yes

4. Is the manuscript presented in an intelligible fashion and written in standard English?

Reviewer #1: No

5. Review Comments to the Author

Reviewer #1: This is a good study with unique population that does not get enough attention in the literature. However, the manuscript has lack of strong background information, several methodological weaknesses, and organization issues that need to be improved. See the the attached document for specific recommendations.

6. PLOS authors have the option to publish the peer review history of their article (what does this mean?). If published, this will include your full peer review and any attached files.

Reviewer #1: **Yes: **Elsa M. Orellano-Colon

---

## [Author Response · Author response to Decision Letter 0]

9 Oct 2020

We would like to thank you for handling our manuscript. We appreciate the comments and suggestions of all reviewers and respond to all of them in the document "Response to Reviewers". 

As a result of the corrections and following the suggestions, the new version of the paper is much improved, and we believe it will represent an important contribution to the field.

The editing of the manuscript in relation to the spelling and grammar of the English language was carried out by two co-authors of the article, one who lives in England and has been a professor at the University of Surrey since 2005 and one who is a professor at the Hogeschool van Amsterdam (HvA) and professor at the University of Amsterdam (UvA) at the Institute for Informatics.

In addition, we removed the copyrighted images.

The editor asked some questions that we answered in the documents submitted.

We explainned about the participants' confidentiality and about the entire study having received approval from the ethics committee.

We also attached the .tex file from Latex.

---

## [Editor Report · Decision Letter 1]

29 Oct 2020

Tailoring Digital Apps to Support Active Ageing in a Low Income Community

PONE-D-20-15049R1

Dear Dr. Lorenzi,

We’re pleased to inform you that your manuscript has been judged scientifically suitable for publication and will be formally accepted for publication once it meets all outstanding technical requirements.

Kind regards,

Jeffrey Jutai

Academic Editor

PLOS ONE
---

## [Editor Report · Acceptance letter]

2 Dec 2020

PONE-D-20-15049R1 

Tailoring digital apps to support active Ageing in a low income community 

Dear Dr. Lorenzi:

I'm pleased to inform you that your manuscript has been deemed suitable for publication in PLOS ONE. Congratulations! Your manuscript is now with our production department. 

Kind regards, 

on behalf of

Dr. Jeffrey Jutai 

Academic Editor

PLOS ONE